# A Mini Review on Fluid Topology Optimization

**DOI:** 10.3390/ma16186073

**Published:** 2023-09-05

**Authors:** He Li, Cong Wang, Xuyu Zhang, Jie Li, Jianhu Shen, Shiwei Zhou

**Affiliations:** School of Engineering, RMIT University, Melbourne 3001, Australia

**Keywords:** fluid topology optimization, isogeometric analysis, Navier–Stokes flows

## Abstract

Topology optimization holds great potential to achieve the best performance for various fluid-related applications like aircraft components and microfluidic mixers. This paper reviews comprehensively the technical progress of this field over the last decade from the viewpoint of structural expression. The density-based approach has been widely adopted to design structures due to its simple concept, ease of implementation, and robustness. Different designs using such a pointwise method for systems under Stokes, laminar Navier–Stokes, turbulent, non-Newtonian, and steady-state/unsteady-state fluid flows are reviewed and discussed in depth. Both isogeometric analysis and the moving morphable components/voids methods will demonstrate their advantages regarding integration with computer-aided design. The moving morphable components/voids method also significantly reduces computing costs. From the viewpoint of boundary smoothness, we are most concerned about whether smoother boundaries can reduce objective functions such as energy dissipation even lower. Therefore, this work also concentrates on level set and spline expression methods. Furthermore, we identify isogeometric analysis and machine learning in shaping the field’s future. In addition, the review highlights the following two challenges: achieving accurate fluid model construction and the relatively limited experimental validation of fluid topology optimization outcomes.

## 1. Introduction

Since the seminal work on homogenization proposed by Bendsoe and Kikuchi in 1988 [1], structural topology optimization has attracted broad interest in academic and engineering communities thanks to its powerful versatility in designing various structures in different mechanical and physical scenarios. Compared with the traditional trial-and-error techniques, this systematical approach significantly improves design efficiency and obtains better performances. Topology optimization aims to improve structures’ mechanical performances and materials’ physical properties under various constraints. During optimization, structural topology naturally changes due to the emergence and merging of void phases. As such, it offers the flexibility to explore unconventional and innovative geometrical configurations that may not be apparent through traditional approaches, such as size and shape optimization.

One of its stunning achievements, like lightweight structures [2], is the performance enhancement of fluid systems governed by various fluidic models. Specifically, it allows fluid to flow through the channels with minimal drag/lift coefficients, minimal pressure drop, and the most efficient heat transfer rate. Tremendous efforts have been put into this field to explore its diverse applications in aerospace, automotive, energy, and biomedical areas, such as aircraft components [3,4,5], microfluidic mixers [6,7,8], and heat exchangers [9,10,11]. Thus, a compressive review is demanded to summarize the recent progress of fluid structural topology optimization, especially from the viewpoint of geometry expression, and to overview its future development with recently developed optimization approaches.

The remainder of this paper is organized as follows. Section 2 and Section 3 introduce fluid topology optimization from the aspect of geometrical expression, including density-based method, moving morphable components/voids, level-set, and non-uniform rational B-splines (NURBS) methods. Isogeometric analysis and machine learning methods are also addressed when predicting future trends in this field. Furthermore, a brief explanation of the rationale of each technique is provided to link their application in fluid topology optimization. Section 4 contains a comprehensive discussion by combining all the methods, providing their strengths and weaknesses. Moreover, a table summarizing the research using the two main fluid topology optimization methods is provided. The concluding part of the article presents the following two potential future challenges: enhancing the precision of modeling complex fluid systems and conducting additional experimental validation of fluid topology optimization.

## 2. Explicit Methods

### 2.1. Density-Based Method

Topology optimization using a density-based method is a powerful approach to optimize the material distribution within a given design space to achieve the desired structural performance. The fundamental idea behind density-based methods is to represent the design domain using a mesh and to assign artificial density values based on the material occupying ratio within an element. The optimization process involves iteratively adjusting these density values to find the optimal distribution that meets specified objectives while satisfying design constraints. These objectives could include minimizing compliance, maximizing stiffness, reducing stress concentration, or achieving other performance criteria. The optimal artificial density ranging from 1 (solid) to 0 (void) allows for the elimination of inefficient materials, as shown in Figure 1 [2].

The density-based method is one of the most extensively utilized techniques in fluid dynamics. Its incorporation into fluid topology optimization dates back to the early 21st century, marking a pivotal development in the field. The seminal paper by Borrvall and Petersson first solved the topology optimization problems for fluid systems [12]. The optimization problem is framed as minimizing the total potential power to achieve the maximum flow rate in the fluid domain without considering body force. They give proof of convergence and show that the method does not require regularization.

The governing equations commonly used in density-based fluid topology optimization problems are the laminar Navier–Stokes equations, which can be characterized as follows [13]:(1)$$\rho \frac{\partial u}{\partial t}+\rho \left(u\cdot \nabla \right)u+\alpha u=-\nabla p+\nabla \cdot (\mu \nabla u)+f\;$$
(2)$$\frac{\partial \rho }{\partial t}+\nabla \cdot (\rho u)=0$$
where *ρ* is the fluid density, **u** indicates the fluid flow velocity, *µ* represents the fluid dynamic viscosity, *p* denotes the pressure, **f** symbolizes the external body forces, and *α* is the inverse permeability. Based on lubrication theory, this method formed a fictitious fluid/solid domain by adding a friction term *α***u** to the equation with the inverse permeability *α*, shown in Figure 2. For the steady-state problem, this equation can be transformed into a time-independent equation by removing the first term at the left of the equations. For the Stokes flow problem, the inertia term can also be neglected because it is much smaller than the viscous forces.

Two years later, Gersborg-Hansen et al., extending the previous work, achieved optimal design of the flow channel topology for two-dimensional constant incompressible laminar viscous flow, steady-state Navier–Stokes model, which makes the governing equation change from linear to nonlinear [13]. Later, Guest and Prévost proposed a new methodology by using the Darcy–Stokes model to optimize the layout, in which existing stabilized mixed finite element methods are used to deal with coupled flow problems. The solid phase was treated as a porous media, while the flow through solid material was captured using Darcy’s law to regularize the optimization problem [14]. Then, Olesen et al. presented a versatile high-level programming-language implementation to take Navier–Stokes flow topology optimization design in a commercial software package based on the finite element method FEMLAB, which minimized the required programming effort and significantly contributed to the development and application of fluid topology optimization [15]. Next, the Lattice Boltzmann method, a discretization method, was used by Pingen, Evgrafov, and Maute for solving 2D and 3D fluid topology optimization problems as an approximation of Navier–Stokes flow [16]. Based on the professional finite volume computational fluid dynamics toolbox OpenFOAM, two numerical examples of topology optimization of Navier–Stokes flows were given by Othmer [17]. Pereira et al. published a polygonal finite element-based MATLAB frame for topology optimization of dissipative power minimization problem in Stokes flow [18]. The main MATLAB functions in this frame can be obtained according to their previous work [19,20] by changing a few lines of the code.

Kreissl, Pingen, and Maute first proposed a method to optimize unsteady flow problems, indicating that its results differ significantly from steady-state systems. They employed a stabilization technique to effectively address unsteadiness, effectively mitigating velocity oscillations within porous materials. Notably, they emphasized that while the Brinkman method effectively partitions the domain into fluid and solid regions, it only partially prevents pressure diffusion through the solid phase, potentially impacting overall predictive accuracy [21]. Extending the exploration into unsteady fluid flow, Nørgaard et al. harnessed the lattice Boltzmann method to simulate unsteady incompressible flows in topology optimization [22]. In a parallel endeavor, Nobis et al. explored unsteady Navier–Stokes flows in topology optimization using a high-order spectral element method. This method attains precision comparable to traditional finite element methods and achieves this precision with fewer degrees of freedom, marking an efficiency improvement [23]. 

Yoon developed a new finite element-based turbulence topology optimization using the Spalart–Allmaras equation [24]. Dilgen et al. used the one-equation Spalart–Allmaras model and the two-equation k-ω model to optimize turbulent flows, as shown in Figure 3 [25]. Wu and Zhang applied topology optimization to the aerodynamic design under turbulent flow and optimized topology for a low-drag profile in turbulent flow using a modified Launder–Sharma k-ω model [26].

As for non-Newtonian fluids, Pingen and Maute studied topological optimization problems for non-Newtonian fluids based on the Lattice Boltzmann method. They also presented an outlook on topological optimization of non-Newtonian fluids, including applications to biomedicine and different non-Newtonian fluid properties [27]. Suárez et al. solved non-Newtonian fluid topology optimization problems using the virtual element method and posted the freely available code of it. By comparing the finite element and virtual element methods, they demonstrated that they had better computational performance and were well-suited for topological optimization problems in more complex domains. They investigated using the virtual element method to solve high Reynolds number problems [28].

Some experimental validations were based on the density method of fluid topology optimization and computational fluid dynamics. Both Lin et al. and Lim et al. studied fixed-geometry fluid diodes and conducted experimental validation of this design using 3D printing, which verified that the structure possesses the characteristics of a fluid diode after topology optimization [29,30]. Sá et al. performed multi-objective optimization on a small-scale pump, aiming to minimize energy dissipation and vorticity, and validated the results through experimental testing using 3D printing [31]. Zhou et al. also used a density model based on Darcy interpolation for topology optimization of fluid diodes and first fabricated valveless micro-pumps based on fluid diodes through microelectromechanical systems technology [32]. The density-based method is the most used method applied in commercial and experimental validation due to its ability to provide efficient computations and manufacturable designs, meeting the requirements of practical engineering applications.

### 2.2. Moving Morphable Components/Voids Method

The moving morphable components topology optimization approach was initially introduced by Guo et al. in 2014 [33]. Subsequently, they proposed the codes of this method [34] and extended it to 3D and multiple materials topology optimization problems [35,36]. This methodology aims to create adaptable structures by optimizing the arrangement and configuration of components that can undergo deformations or movements in response to changing conditions. The method comprises several distinct structural components, the basic building blocks for topology optimization. Each of these components occupies a specific volume within the design domain. The proposed framework allows components to overlap, leading to changes and optimizations in the structure’s layout. Figure 4 shows the optimization process of this method, which performs topology optimization by calculating component length, thickness, and inclined angle. Based on the moving morphable components method, they also developed the moving morphable voids method and introduced the B-spline to represent the smooth structural boundaries [37]. B-spline is a mathematical curve or surface representation technique widely used in computer-aided design. It provides flexibility in defining complex shapes through control points and basis functions, making it suitable for accurately describing geometries and optimizing structural designs. Further details about this are discussed in Section 3.2.

Researchers focus more on fluid-related thermal systems than purely fluid structures in this method. Yu et al. first introduced the moving morphable components method to thermal-fluid problems, in which the objective is minimizing heat compliance and fluid power dissipation. Because cooling device pipes often exhibit larger curvature than bearing structure, a higher-order component is utilized, and elliptic joints are introduced to enhance the connections between high-order components. Additionally, they developed a feature size control technique to limit the minimum width of pipes based on manufacturing capacity and removed unnecessary components. They compared the results with those obtained using the density-based method, revealing differences in the optimization processes of the two optimal structures. The density-based method initially establishes primary fluid pathways. As the optimization progresses, additional thin channels emerge to link these main paths. The results obtained using the moving morphable components method, on the other hand, exhibit relatively stable evolution processes due to the initial components setup. Additionally, the feature size control technique prevents the formation of those thinner channels [38]. 

Their team employed a mixed topology optimization approach based on the framework to perform thermal-fluid topology optimization problems. They treated the cooling channels with the density-based method and considered the heat sources as moving components. Also, they proposed an exchange algorithm redistributing heat intensities during optimization to reduce the impact of the initial distribution of heat sources [39]. As for the moving morphable voids method, Liang et al. employed it to optimize thermal-fluid regions and introduce isogeometric analysis to solve steady-state Navier–Stokes and heat equations. The objective was to maximize heat exchange performance under pressure drop constraints and ensure manufacturability [40].

## 3. Implicit Methods

### 3.1. The Level Set Method

Besides the density-based approach, the level set method, as a classical method, has also been widely used for topology optimization, starting with this seminal article by Wang et al. [41]. Topology optimization using the level set method is versatile for optimizing complex shapes and structures. The level set function assigns signed distance values to points within the design domain, representing their distances from the interface. The optimization process involves changing the shape of the zero-level contour to achieve the desired design objectives. Specifically, the zero-level contour and the negative and positive values in the level set function represent separately structural boundaries, voids, and solids in the design domain. By manipulating the values of the level set function, the position of the interface between the design domains can be changed, thus achieving topological optimization of the structure [42], shown in Figure 5. Due to that, interfaces are captured using the zero-level contour, providing a clear and well-defined representation of different regions in this approach.

Duan et al. first used a variational level set method to solve Navier–Stokes flow topology optimization [44,45,46]. In a parallel exploration, Zhou and Li demonstrated 2D and 3D examples via the level set method in Navier–Stokes flow topology optimization. A zero-level contour of a higher-level scalar function and variational calculus were utilized separately to express the solid/fluid interface and derive the normal velocity of the level set function, shown in Figure 6. In addition, a bisection method was introduced to overcome the gradual failure volume constraint due to errors [47]. In pursuit of accurate and computationally efficient fluid flow simulations, Challis and Guest introduced a level set algorithm tailored for the Stokes flow topology optimization problem. This method offers a notable advantage by enabling the direct application of the no-slip boundary condition during each iteration. By confining computations solely to the fluid portion of the domain, this method ensures more accurate modeling and significantly reduces computational costs, distinguishing it from density-based approaches [48]. Addressing convergence challenges posed by higher Reynolds numbers in flow problems, Pingen et al. adopted a multifaceted approach. They combined a hydrodynamic lattice Boltzmann method with a parametric level set approach. This innovative strategy involved discretizing the level set function using radial basis functions, leading to optimization variables directly influencing the outcome [49]. Deng et al. demonstrated topological optimization of 2D and 3D Navier–Stokes fluids with body forces using an implicit variational level set method [50]. In the same year, Deng et al. used a variational level set method for topological optimization of unsteady Navier–Stokes fluids with or without body forces [51]. 

Kreissl, Pingen, and Maute combined the Lattice Boltzmann method with the level set method for shape optimization of fluids [52]. Based on this, Kreissl and Maute developed a level set method to obtain the fluid–solid boundary, while the extended finite element method was utilized as the discretization method. Since the proposed approach does not depend on fictitious porous materials, there is no need to establish a material interpolation scheme or impose specific objectives and constraints to achieve a binary material distribution. As a result, the proposed topology optimization method is highly adaptable and can be flexibly applied to solve optimization problems with various objectives and constraints [53]. Further explorations delved into fluid-structure interaction issues. By recognizing the significance of extended finite element and horizontal basis methods, Jenkins and Maute applied them to topological optimization problems in fluid-structure interactions [54]. Duan, Qin, and Li proposed a new algorithm for Stokes flow topology optimization named implicitly coupled level set methods. Due to the maturity of material distribution methods in solid topology optimization, they determined the local permeability by combining the level set method and material distribution information rather than explicitly evolving the boundary [55]. Later, they developed an adaptive level set method for topology optimization of the Navier–Stokes flows [56]. At the same time, they also published density-based fluid topology optimization based on an adaptive mesh [57].

Sasaki et al. used the moving particle semi-implicit and level set method to optimize fluid topology. The algorithm demonstrates that the optimized structure is consistent with that obtained by the finite element method [58]. Nguyen et al. performed topological optimization of transient flows under oscillation using level set and the lattice Boltzmann method [59]. Concurrently, Duan, Dang, and Lu expanded upon their earlier work [44,45,46] to address Navier–Stokes fluid problems. Their novel approach aimed to accurately portray optimal shape representation while achieving significant computational efficiency gains [60]. 

Additionally, blood vessels are a critical component of many biological systems, and the presence of non-Newtonian fluid behavior within these vessels adds a layer of complexity to fluid flow dynamics. Some critically crucial vascular fluid studies have already been published [61,62,63,64], yet the topology optimization in this field still needs to be improved. Only Zhang and Liu have studied the arterial bypass design problem using level-set topology optimization [65]. Understanding and optimizing these interactions through fluid topology optimization can have significant implications for improving medical devices and treatments.

### 3.2. NURBS Method

In this approach, spline functions, which are mathematical curves that can be smoothly adjusted, are employed to define the shape and topology of the design. On the other hand, the traditional linear shape functions of the finite element method are replaced with higher-order NURBS basis functions used in the isogeometric analysis, allowing a more accurate representation of complex geometries. The fundamental idea is to use the same geometric description for analysis as used in computer-aided design models, promoting seamless data exchange between design and analysis. Figure 7 illustrates an example to demonstrate how isogeometric analysis functions as a link between computer-aided design and computer-aided engineering. Combining isogeometric analysis with density-based isogeometric topology optimization [66,67,68], level set isogeometric topology optimization [69,70], and moving morphable components/voids, is a new trend of topology optimization [71,72]. In addition, Seo et al. initially applied isogeometric topology optimization using trimmed spline surfaces. Spline surfaces and trimming curves depict geometric design models’ external and internal boundaries. In this method, the design variables in topology optimization consist of the control point coordinates of the spline surface and the trimming curves [73].

However, isogeometric analysis for fluid topology optimization is still rare. To date, only a few fluid-related isogeometric shape optimizations have been published. Based on the trimming isogeometric analysis [73], Park et al. took the control point as a design variable with trimming curves to perform the shape optimization of the Stokes fluid problem to minimize drag forces and functional dissipation. Due to that, the smooth boundaries of the analysis model are achieved through spline functions in isogeometric analysis. They highlighted that this property allows for the efficient treatment of design-dependent load problems, such as convection and pressure [74]. Later, Nørtoft and Gravesen also used the method with control points as design variables for shape optimization of the incompressible Navier–Stokes problem [75]. Figure 8 illustrates how the shape optimization is performed. It takes the coordinates of control points on the boundary as design variables. During the optimization process, as the shape of the domain changes, the internal parametrization must be thoroughly updated. Internal control points are linked to the design variable control points to avoid self-intersections and ensure better uniformity [76]. Chen et al. also used the control points on the boundaries as design variables to optimize the fluid cooling channel shape, simplifying the Navier–Stokes equations according to Darcy’s law. By comparing the results by finite element analysis under the same model, it can obtain a higher accuracy solution with fewer elements. Their findings underscored that adopting isogeometric analysis yielded smoother and more continuous boundaries, leading to a marked reduction in computational cycles compared to alternative methods [77].

### 3.3. Machine Learning Method

Machine learning has shown great potential in various fields, which focuses on utilizing data using statistical methods to learn meaningful patterns. Deep learning has also been widely used as a subset of machine learning. The human brain’s neural structure inspires it and trains hierarchical neural networks using the data to enhance learning [78]. Because of the increased computational costs resulting from the extensive iterations in conventional topology optimization methods, machine learning topology optimization has emerged to reduce computational costs. Despite its relatively short development time, machine learning topology optimization has already found widespread application in solid mechanics [79,80,81]. However, machine learning has few applications in fluid topology optimization, like the isogeometric analysis method. 

Gaymann and Montomoli introduced the concept of making the fluid-structure topology optimization problem gamification. They proposed a method for optimizing the fluid-structure to minimize pressure difference using deep neural networks and Monte Carlo tree search. In this optimization approach, the deep neural networks determine the winner and decide whether the game ends while predicting the pressure difference. The player’s future movements are determined through iterative Monte Carlo searches based on these outcomes, achieving the final optimized shape [82].

Ban and Yamazaki used clustering/classification methods and exploration strategies to solve the optimization problems of discontinuities, infeasible regions, and high-dimensional optimization problems in advanced global topological optimization. Combining these methods, they used machine learning technologies to publish the black-box function for the aerodynamic topology optimization problem. By only using the objective function and constraints, the newly developed design tool can perform effective fluid topology optimization [83]. Hammond, Pietropaoli, and Montomoli used data-driven modeling for turbulence models to perform topology optimization of backward-facing step and u-bend separately to minimize pressure drop. They highlighted that, by being data-driven, the optimization solver could replicate the accuracy of high-fidelity simulations while keeping the computational cost low [84]. Furrokh and Zhang used stochastic gradient descent methods, a fundament for optimizing deep neural networks, in piston channel topology optimization. It was the first-time stochastic mechanisms were used to express fluid interfaces. Comparing the gradient and stochastic gradient descent methods shows that the stochastic approach has proven effective in optimizing the flow-solid surface for nonconvex problems. In more detail, the stochastic gradient descent method can obtain smaller objective function values and avoid the false minima in the gradient descent method [85]. 

Deng et al. presented a novel approach called self-directed online learning optimization, which combines deep neural networks with finite element method calculations. Deep neural networks are employed to learn and replace the objective function regarding design variables. The deep neural networks adjust to the evolving training data, leading to improved predictions within the targeted region until convergence [86]. This approach not only addresses fluid topology optimization problems but also encompasses the optimization of heat transfer and 3D truss structures. Additionally, it demonstrates a remarkable reduction in computational time, exhibiting superior performance in comparison to a recent algorithm [82]. Overall, using machine learning in fluid topology optimization can significantly improve the efficiency and accuracy of the optimization process and enable the solving of more complex and challenging problems.

## 4. Comparisons and Discussion

Table 1 encompasses most research on this level set and density-based methods, which have demonstrated effectiveness and applicability to various fluid models. The density-based method utilizes a density field or material distribution using material interpolation equations to define different segments within the design domain. However, artificial densities cannot perfectly represent a binary distribution, thereby affecting the calculation of the objective function value. The density-based method may yield objective function values, such as dissipated energy, lower than the analytical solution of the optimized fluid structure. This discrepancy arises from intermediate density elements at the fluid–solid boundaries, leading to increased actual volume and reduced energy dissipation. However, despite the difference between the objective function values and the analytical solution, it is considered acceptable with the optimized design [12]. In contrast, the level set method, which does not require an additional material distribution equation, ensures clear boundaries in the fluid domain throughout the optimization process without intermediate density elements. Additionally, the sensitivity to the initial density may result in variations in the optimized fluid domain for different initial artificial densities. Thus, selecting an appropriate initial density distribution is crucial for influencing the design of the optimized structure and achieving the desired optimization outcomes. The level set method handles interface jumps within the level set function regarding interface reconstruction. That is why the outcomes of most topology optimization methods based on the level set approach typically avoid mesh dependency. In contrast, the density-based method requires additional techniques, such as filtering, to ensure smooth and coherent interfaces. The level set method may require higher-order numerical techniques to handle interfaces during computations, while the density-based method commonly relies on more uncomplicated numerical strategies. As a result, the density-based method typically offers higher computational efficiency. The level set method often incurs higher computational costs. Also, the convergence speed depends on several factors, including the parameterization of the level set function, the optimization problem formulation, the choice of updating information, the updating process, and regularization methods [42].

Although density-based and level set methods have been well developed, their topology optimization results often need help to integrate effectively with computer-aided design systems. When dealing with 3D problems, the increased number of design variables also leads to higher computational costs. As a result, the moving morphable components/voids method was proposed. Compared to the previous two methods, the method allows for better integration of topology optimization with computer-aided design systems, resulting in a more geometrically explicit and flexible process. This approach employs the geometric information of components as design variables, and the design domain can be composed of a small number of components, thus significantly reducing the number of design variables [33]. Nevertheless, the initial design of components can potentially impact efficiency and accuracy, and components’ initial distribution and shape often rely on the designer’s expertise. Taking the reference [38], for instance, when the initial number of components varies, the resulting topology of fluid channels may also differ. Furthermore, with the advancement of isogeometric analysis, topology optimization and computer-aided design integration have become closer. By combining isogeometric analysis with the density-based and level set methods, although some post-processing is still required, the optimized results can be represented using splines and control points. However, isogeometric analysis currently functions primarily as a numerical method in topology optimization, and the potential of utilizing changes in control point positions to alter both the structural shape and topology have yet to be fully explored. Both the moving morphable components/voids and isogeometric analysis methods have significant advantages in integrating with computer-aided design and reducing computational costs, and further research is needed, particularly in fluid topology optimization.

Since the optimization outcomes are predicted based on pixel similarity, there might be a lack of connectivity, leading to less manufacturability and lower resolution of the topology optimization results. For enhancing manufacturability, this method [38] can be considered by imposing constraints on the minimum width of channels to prevent them from becoming too thin or breaking. However, additional post-processing may diminish the computational efficiency advantage of this approach. 

## 5. Conclusions

Fluid topology optimization identifies the most advantageous fluid flow regions that enhance metrics like drag, lift coefficients, pressure drop, and heat transfer. Its core aim is to craft fluid shapes aligning with designated engineering criteria, improving the system’s effectiveness. It has been developed for two decades and has shown significant potential for future advancements and applications in various industries. 

This paper briefly outlines the pros and cons of recent optimization methods for fluid-structure design. For instance, density-based approaches are computationally efficient and easily shape-controllable, while level-set techniques ensure clear, well-defined fluid–solid interfaces throughout the optimization. Due to its ability to combine with computer-aided design and significantly reduce computational costs, the moving morphable components/voids method also represents an alternative direction. Additionally, integrating isogeometric analysis can enhance geometric representation and optimization accuracy, while machine learning techniques bring data-driven insights and predictive capabilities to the forefront of fluid topology optimization.

Because of the diversity and complexity of fluid flow models, one of the critical challenges in fluid topology optimization is the need to model fluid flow accurately. Given that much research is now directed at dealing with 2D simple fluid topology optimization problems, more complicated fluid patterns and 3D meshing techniques must be developed to simulate fluid flow more accurately although they are computationally expensive, time-consuming, and require more specialized skills and knowledge. Existing reports using simplified models [87,88,89,90] and the use of parallel computing [91,92], do not fully simulate more complicated fluid problems. The computational cost remains a challenging issue for the ordinary laptop. 

Using 3D printing technology, the experimental validation methods in fluid topology optimization include flow visualization, pressure drop measurements, and computational fluid dynamics simulations [29,30,31,32]. These approaches can provide valuable insights into the practical benefits of the optimized design, assess the fluid system’s flow characteristics, and inspire engineers of fluid structures. However, a few studies have involved experimental validation, and there is still a considerable distance in applying fluid topology optimization techniques in reality.

## Figures and Tables

**Figure 1 materials-16-06073-f001:**
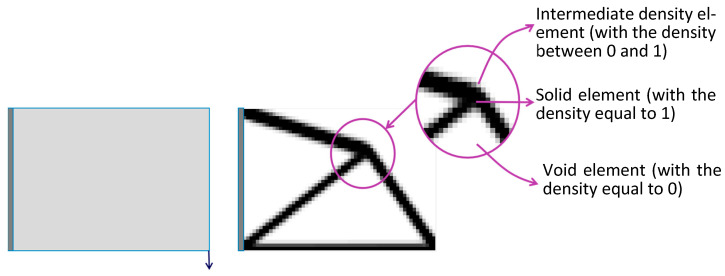
Topology optimization of a cantilever beam with a 20% volume fraction.

**Figure 2 materials-16-06073-f002:**
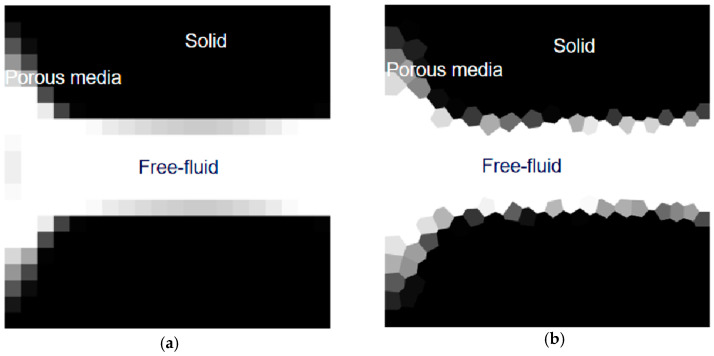
Density-based fluid–solid expression of a diffuser example using (**a**) rectangular elements; and (**b**) polygonal elements.

**Figure 3 materials-16-06073-f003:**
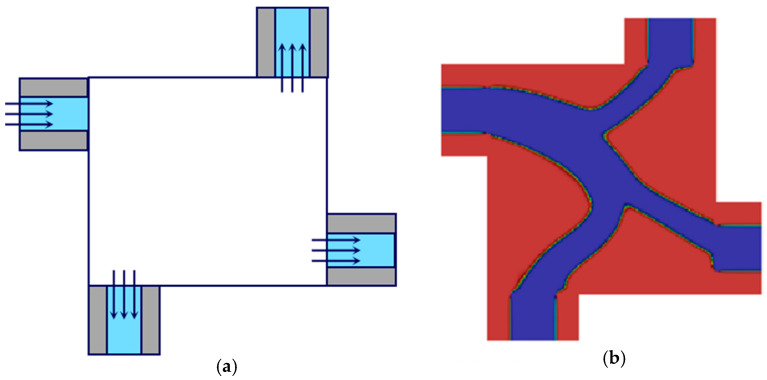
(**a**) Schematic illustration of the 2D flow manifold problem and (**b**) topology optimization for Re = 3500 using the k–ω turbulence model. (Reproduced with the permission from Ref. [25]. © 2017 Elsevier B.V. All rights reserved).

**Figure 4 materials-16-06073-f004:**
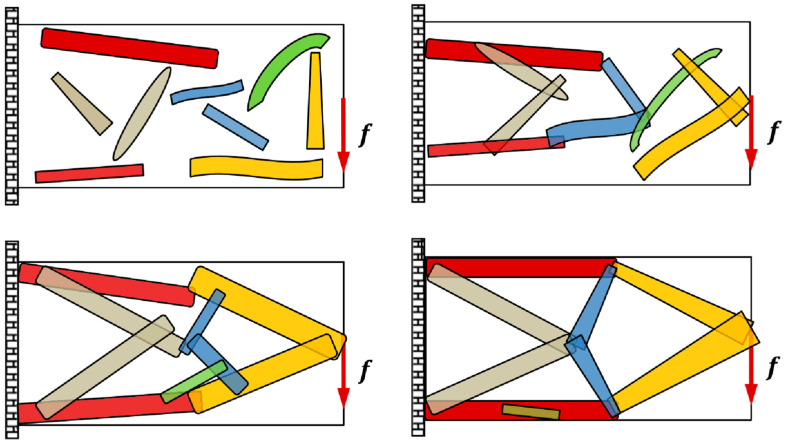
The basic idea of the moving morphable components-based topology optimization approach (Reproduced with the permission from Ref. [36]. Copyright © 2017 John Wiley & Sons, Ltd.).

**Figure 5 materials-16-06073-f005:**
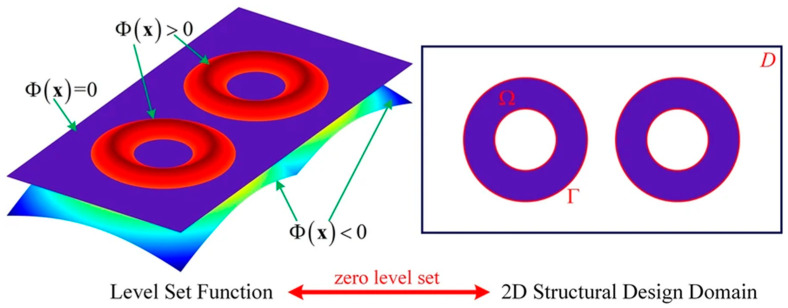
A 3D level set function and its zero-level contour of a 2D shape (Reproduced with the permission from Ref. [43]. Adapted from Gao et al. (2020) under the Creative Commons CC BY license).

**Figure 6 materials-16-06073-f006:**
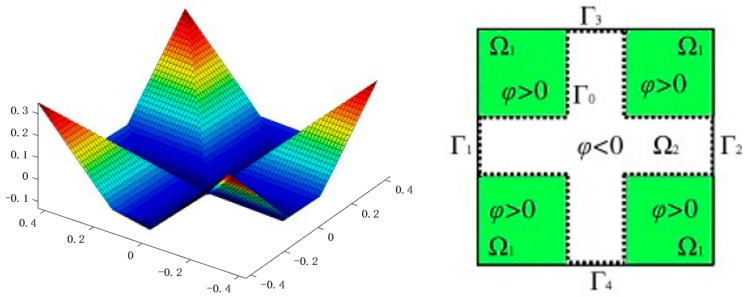
Expression using the level set method. (Reproduced with the permission from Ref. [47]. Crown copyright © 2008 Published by Elsevier Inc. All rights reserved).

**Figure 7 materials-16-06073-f007:**
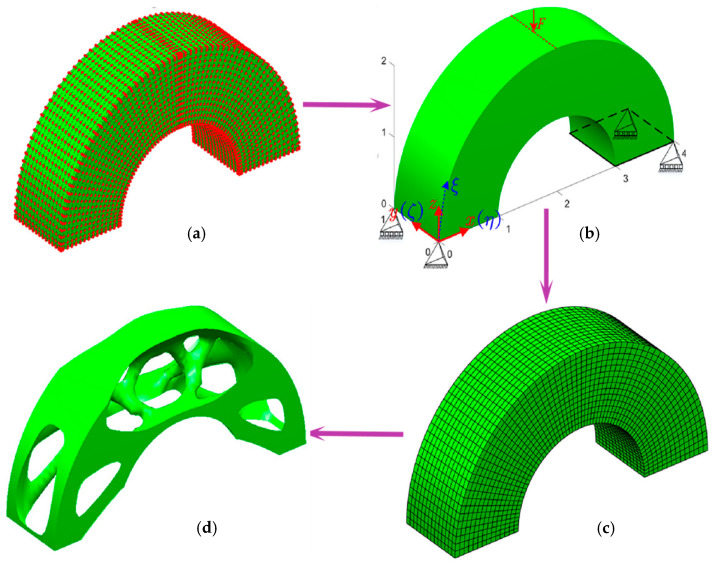
NURBS based optimization for a 3D bridge: (**a**) NURBS solid modal; (**b**) structural design domain; (**c**) isogeometric analysis modal; (**d**) topology optimization result (Reproduced with the permission from Ref. [67]. © 2019 John Wiley & Sons, Ltd.).

**Figure 8 materials-16-06073-f008:**
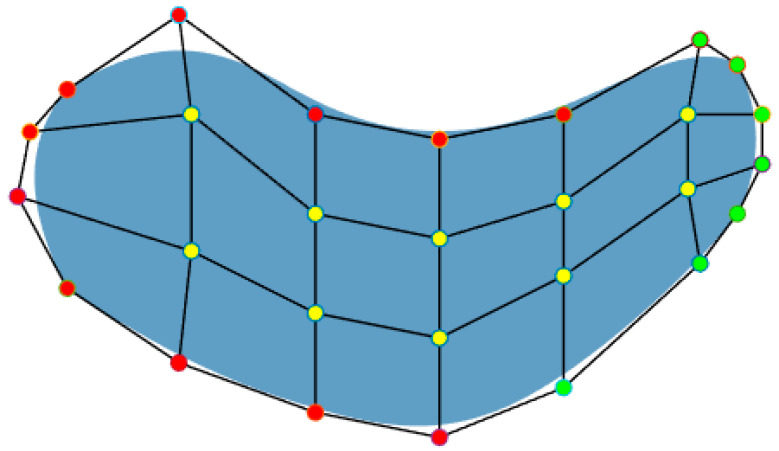
A common expression of optimization problems using the NURBS (design control points (green), linked control points (yellow), and fixed control points (red)).

**Table 1 materials-16-06073-t001:** Summary on fluid topology optimization methods.

Article	Year	Objective Function (Extremization)	Solver	Method	Fluid Type
Duan et al. [45]	2008	Dissipated power	FEM	Level set	Steady Navier–Stokes
Zhou & Li [47]	2008	Dissipated power	FEM	Level set	Steady Navier–Stokes
Challis & Guest [48]	2009	Dissipated power	FEM	Level set	Stokes
Pingen et al. [49]	2010	Pressure drop	LBM	Level set	Steady Navier–Stokes
Jenkins and Maute [54]	2015	Strain energy augmented by a perimeter penalty	XFEM	Level set	Steady Navier–Stokes
Zhang and Liu [65]	2015	The integral of the squared shear rate	FEM	Level set	non-Newtonian
Pereira et al. [18]	2016	Local velocity/Dissipated power	Polygonal element method	Density	Stokes
Duan et al. [55]	2016	Dissipated power	FEM	Coupled Level set	Stokes
Yoon [24]	2016	Dissipated power	FEM	Density	The Reynolds- Averaged Navier–Stokes
Dilgen et al. [25]	2017	Dissipated power	FVM	Density	The Reynolds- Averaged Navier–Stokes
Sasaki et al. [58]	2019	Pressure drop	FEM	Level set	Steady Navier–Stokes
Nguyen et al. [59]	2020	Dissipated kinetic energy	LBM	Level set	Unsteady Navier–Stokes
Nobis et al. [23]	2022	Dissipated power	Spectral element method	Density	Steady/Unsteady Navier–Stokes
Suárez et al. [28]	2022	Dissipated power	Virtual element method	Density	non-Newtonian

## Data Availability

Data will be made available on request.

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
