# Peer review of "A Mini Review on Fluid Topology Optimization"

_materials, 2023, doi:10.3390/ma16186073_

Round 1

Reviewer 1 Report

The following suggestions must be incorporated for improving of the manuscript, the decision on acceptances of the paper is primarily depends on the incorporation of the following comments in the revised version of the manuscript.

1.      The abstract should be revised, it should be presented in a way to reflect your research work, connect the title to the abstract and abstract to introduction

2.      In the Introduction, the authors need to describe not only what was done in previous works, i.e. it is also necessary to stress the main  results of each of them, otherwise the reader will not properly understand how the present work compares to the state-of-the-art on the subject. Once the state-of-the-art is characterized, the authors should state the objectives of their work based on what lacks to be done in the subject, emphasizing its novelty and possible answers to open questions. All references should be cited in uniform order.

3.      clarify and discuss the novelty and the significance of the results obtained here, and compare them with those available in the literature, also including discussions on potential applications;

4.      Is the method applied here is new? Justify and include details

5.      There are problems with sentence structure, verb tense, and clause construction

6.      Tables and basic equations are presented without details and references

7.      The conclusion section needs revision. Authors should revise the conclusion section to provide conclusive statements on the research questions posed at the end of the introduction. Try to itemize all the conclusive facts.

  Serious improvement in English is needed for the entire article.

Reviewer 2 Report

In the introduction, the authors should aim to present a comprehensive overview of fluid topology optimization problems, clearly outlining the key processes such as the forward and backward solve, and the necessity for optimization techniques. As it stands, the paper dives into details too abruptly, which can be daunting for those unfamiliar with the field. For instance, delving into SUPG stabilization without first providing the necessary context creates an context that can leave readers floundering. The paper would benefit from presenting a broader overview of the field before narrowing down to specificities. 

At present, the paper reads more like an extremely brief summary of existing research rather than an accessible and engaging introduction to the field. There is an overreliance on cited papers with minimal explanation. Although referencing other papers is crucial, they should be woven into a comprehensive narrative that gives novices in the field a clear understanding of the subject matter. It's important to remember that the primary audience for review papers isn't experts already well-versed in the field. 

The ultimate goal should be to create a meaningful and engaging survey of the field. Although absolute completeness is not a requirement, the paper should not sacrifice clarity. The beginning of section 2.3 is a good example of introducing the topic appropriately and should be emulated throughout the paper.

While I understand that the authors have intended for this to be a "mini-review," in its current state, it resembles rough notes from a PhD student rather than a rigorously composed scholarly paper. A mini-review, although not expected to be exhaustive, should still paint a coherent and clear picture of the field. Regrettably, the current manuscript is neither sufficiently clear nor adequately comprehensive.

Other minor notes:

Regarding formatting, please ensure consistency, especially in terms of spacing between reference numbers and the text. 

The sentence, "The function PolyTop, from their previous work [9, 10], can only change a few lines of code in this frame," is confusing and requires clarification. 

The English is clear with minor issues.

Reviewer 3 Report

In their review study, the authors reviewed recent development in fluid–structure topology optimization. This study is on an important topic, but I would suggest referring to the questions and comments below to make improvements and resubmit:

1-      The introduction section should be improved to show the importance of this topic and defined the fluid-structure topology optimization methods.

2-      The differences between the methods should be clearly stated. The comparison section should be improved using more previous studies.

3-      Section 2.4 should be extended using up-to-date references.

4-      References 56-59 can be added to the literature review section.

Minor editing of English language required. Some grammatical mistakes and typos should be reviewed and improved.

Reviewer 4 Report

The field would benefit from a more in-depth review as opposed to a mini review.

The abstract should highlight some of the major discussions in the manuscript as opposed to what the author looks forward to in terms of future work.

The introduction and background should bullet point some of the major topics of discussion.

The manuscript should include higher quality images.

Fluid topology extends to many areas of fluid dynamics.  Its surprising that the work doesn't discuss laminar and turbulent modelling in cardiovascular flow.  Some to include are:

Hewlin, R.L., Jr.; Tindall, J.M. Computational Assessment of Magnetic Nanoparticle Targeting Efficiency in a Simplified Circle of Willis Arterial Model. Int. J. Mol. Sci. 2023, 24, 2545. https://doi.org/10.3390/ijms24032545

Kumar, N., et al. (2023). "Advances in the application of computational fluid dynamics in cardiovascular flow." Cogent Engineering 10(1): 2178367.

Li, G., Wang, H., Zhang, M. et al. Prediction of 3D Cardiovascular hemodynamics before and after coronary artery bypass surgery via deep learning. Commun Biol 4, 99 (2021). https://doi.org/10.1038/s42003-020-01638-1

Stanley, N., Ciero, A., Timms, W., and Hewlin, R. L., Jr.A 3-D Printed Optically Clear Rigid Diseased Carotid Bifurcation Arterial Mock Vessel Model for Particle Image Velocimetry Analysis in Pulsatile Flow ASME Open J. Engineering ASME. January 2023 2 021010 doi: https://doi.org/10.1115/1.4056639

The abstract needs to provide a more in-depth summary.  Perhaps including bullet point discussions of the take-away points.

Minor english issues exists, although I encourage the authors to recheck the manuscript for english issues.

Reviewer 5 Report

The article is potentially attractive to readers interested in gaining some guidance in the field of fluid topology optimization. In principle, the manuscript could be accepted practically as it is. However, the reviewer indicates some points to be addressed.

1) While this is a mini fluid topology optimization review, audiences who are interested in but unfamiliar with topology optimization would benefit from some additional background information. Thus, a brief preliminary explanation of the rationale and differences between density-based and level set conventional methods is most welcome.

2) Two typos were detected on page 4: “Spalart-Allmeras”; “Naiver-Stokes”.

3) Figure 4 was neither mentioned nor discussed. By the way, a brief discussion of each figure presented is recommended.

Round 2

Reviewer 1 Report

accept 

Reviewer 4 Report

none
